# Morphology and calcification characterization in patients undergoing TAVI: A 3D statistical shape modelling study

**Raphaël Sivera**[1,◉,*], **Ebba Montgomery-Liljeroth**[1,◉], **Yaxi Chen**[1], **Silvia Schievano**[1], **Jan Brüning**[2], **Wouter Huberts**[3], **Anthony Mathur**[4,5,6], **Andrew Cook**[1], **Kush Patel**[1,4], **Claudio Capelli**[1]

**1** Institute of Cardiovascular Science, University College London, London, United Kingdom, **2** Institute of Computer-assisted Cardiovascular Medicine, Charité Universitaetsmedizin, Berlin, Germany, **3** Department of Biomedical Engineering, Eindhoven University of Technology, Eindhoven, The Netherlands, **4** Bart's Heart Centre, Barts Health NHS Trust, London, United Kingdom, **5** William Harvey Research Institute, Queen Mary University of London, London, United Kingdom, **6** NIHR Barts Biomedical Research Centre, Queen Mary University of London, London, United Kingdom

◉ These authors contributed equally to this work.
* r.sivera@ucl.ac.uk

**Data availability statement:** Access to de-identified data is available through the SIMCor Virtual Research Environment (VRE) at

## Abstract

Aortic stenosis (AS) is a common valvular disease becoming more prevalent globally due to the aging of the population. Transcatheter aortic valve implantation (TAVI) is a minimally invasive intervention indicated for AS patients as alternative to surgical replacement. TAVI is to date an established procedure. However, it has been often associated with complications such as paravalvular leakage (PVL) or conduction abnormalities. Evidence of associations between morphological features of the aortic root, valve calcification measurements and suboptimal procedural outcomes have been suggested but the analyses were limited by availability and reproducibility of clinical measurements. In this work, we aim to enrich the characterization of AS patients referred for TAVI by analyzing the clinical findings in conjunction with advanced morphological analysis of the implantation site including aortic root, left ventricular outflow tract and 3D calcification patterns. A population of consecutive patients with AS (n = 130) who underwent TAVI at our clinical centre were retrospectively selected for this study. Demographic and clinical measurements were collected before and after TAVI. Pre-operative CT images were used to reconstruct 3D models of patient-specific anatomies. Statistical shape modelling was carried out and outcomes were analyzed in conjunction with clinical outcomes. The 3D modelling of the valve calcification rate matched previous clinical descriptions; including the crescent shapes visible on each leaflet and the higher calcification rate of the non-coronary cusp. Higher calcification rate was found in larger valves together with a positive association between each coronary height and the calcification of their respective leaflet. Sexual dimorphism, on both shape and calcification, was recorded beyond the size differences with straighter aortas and higher calcification rate at the junction between the left and right coronary leaflets for males compared to females. Morphological

https://simcor.unitbv.ro/vre. The platform can be accessed by interested users. Access is granted for R&D purpose only, due to ethical restriction associated with the use of clinical data.

**Funding:** The work of RS, EML, JB, WH, and CC was supported by the European Union's Horizon 2020 project SIMCor (grant agreement No. 101017578). The funders had no role in study design, data collection and analysis, decision to publish, or preparation of the manuscript.

**Competing interests:** The authors have declared that no competing interests exist.

differences were significantly associated (p = 0.005) with PVL assessments based on post-operative echocardiograms. Larger aortas and shorter left coronary sinus were associated with less leakage. The outcome distribution appeared to be directly affected by sexual differences and device design. Female phenotypes, smaller and more conic aortic root, were associated with worse outcome. Different patterns in calcification distribution on the leaflets were identified but the association with outcomes is not conclusive. In the future, the presented morphological characterization of patients with AS could contribute to predict post-TAVI PVL and design and test improved TAVI devices.

## Author summary

With age, the aortic valve can stiffen and calcify. The valve opening narrows leading to a partial obstruction of the left ventricular outflow tract. Estimates suggest that around 9 million patients are affected with aortic valve stenotic disease globally. The valve then need to be surgically repaired or replaced depending on the disease severity and the patient's comorbidities. In this work, we present a description of a population of aortic stenosis patients who underwent transcatheter aortic valve implantation (TAVI) at our clinical centre. Our approach enriches the clinical description and the manual measurements with a complete 3D morphological model of the aortic root and its calcification. We highlighted new morphological correlations with post-operative risk assessments, and in particular, we observed that female phenotypes, smaller and more conic aortic root, were associated with worse outcome in average. The observed average calcification reflected the clinical descriptions including the crescent shapes visible on each leaflet, and the higher calcification rate on the non-coronary cusp, but calcification association with outcomes remains unclear. We believe that the novel 3D description of the valve area and of the calcification pattern presented in this work may help clinicians better characterise the disease presentation and inform futures clinical and simulation studies in order to improve device design and treatment processes to reduce the risk of post-operative complications.

## 1. Background

### 1.1. Aortic stenosis

Aortic stenosis (AS) is a common valvular disorder causing obstruction of the left ventricular outflow tract. Calcific aortic stenosis is mainly caused by solid calcium deposits within the valve cusps, where the valve's leaflets stiffen and the opening narrows, resulting an increased pressure gradient across the valve [1,2]. Global estimates suggest a prevalence of around 9 million patients with aortic stenosis in 2022 [3,4].

Transcatheter aortic valve implantation (TAVI) is an established treatment for aortic valve dysfunction, especially for inoperable or high-risk patients who have multiple comorbidities or are elderly [5,6]. Recent large trials have also shown it to be non-inferior to surgical aortic valve replacement (SAVR) in low-risk patients [6]. Globally, the annual number of TAVI for individuals over 65 years of age has significantly increased over the past decade [7]. In 2018 around 180 000 patients annually could be considered for TAVI in the European Union and Northern America alone, which could increase to 270 000 patients annually if the treatments is further expanded to low-risk patients [8].

Approved TAVI devices include balloon-expandable and self-expanding valves, classified after the method used to expand them inside the designated position in the heart. There are a range of different designs within each of these categories, but the balloon-expandable have a tendency to be shorter than the self expandable devices, creating differences in how each fit the anatomy of the heart [9,10].

## 1.2. TAVI: General outcomes and open questions

A common complication arising from TAVI is post-procedural aortic regurgitation (AR), which can be divided up in the two categories: paravalvular leakage (PVL) where blood flows back around the prosthetic valve, and transvalvular regurgitation (TVR) where blood flows back though an improperly closed prosthetic valve. PVL have been linked with increased mortality both short and long-term [11]. PVL characteristics and frequency differ between the different types of implanted valves, but some important features such as the most common locations are shared [12].

Potential predictive markers of PVL are however difficult to characterise. Morphological features, such as the angulation and curvature of the aortic landing zone, directly affect the device implantation [13] and evidences about morphological predictors of PVL have been reported, linking, for example, the angle between the left ventricular outflow tract (LVOT) and the ascending aorta for self-expandable Medtronic CoreValve implants [14]. Semi-automatic approaches have been suggested to improve the implant sizing decision based on pre-operative shape model [15] but the direct relation between morphological features and post-operative results remains open. High quantity of calcium have been associated with higher risk of PVL [16] but the inter-studies variability regarding the most relevant areas (annulus, leaflets, LVOT, etc.) and the best way to quantify the calcification is significant [17]. The analyses are often conducted using region-based quantification but precise localization may be important, with one study [18] suggesting a strong link between scale of calcification at the annulus and level of PVL while no such link was observed for the calcification at the nadir or commissure. The importance of standardized and interpretable features is reinforced by the fact that the relation with outcome is strongly affected by device type and generation [17].

Another common complication is disturbance to the conductivity arising after TAVI, creating a need for permanent pacemaker to be implanted. Mechanical compression of the device on the LVOT is reported to be a cause of conduction impairment [19], and the ratio LVOT area/annulus area has been shown to have a predictive ability of imminent post-TAVI procedural need for pacemaker in retroactive patient studies [6]. The mechanical disruption of the conduction system can also be aggravated by increased valvular calcification [20].

## 1.3. Introduction to in-silico medicine and Statistical Shape Modelling (SSM)

Continued increase in safety and minimisation of these complications is a major point of interest of both device development and of the assessment process ahead of TAVI-treatment. Understanding the morphology in full could serve to identify risks factor and therefore improve the planning of TAVI procedures. Computational modelling has the potential to describe quantitatively the implantation site and model the possible interactions with devices. As shown in Verstraeten et al. [21], a model of the anatomical variations of TAVI patients can be leveraged to generate the necessary synthetic data to conduct in silico trials.

This study aims to fully characterize the implantation site of a cohort of AS patients referred to TAVI procedure. It aims to provide a novel description of the morphology of

the aorta and of the left ventricle outflow track (LVOT), as well as of the aortic calcification distribution and to understand whether there are morphological features which can be predictors of the consequence of the implantation regarding flow properties and impact on the heart conductive network.

# 2. Results

## 2.1. Population characteristics

**2.1.1. Demographic and clinical information.** The demographics of the included population (n = 130) of patients who underwent TAVI at Bart's Heart Centre (London, UK) and the elementary description of the aortic morphology and function is shown in Table 1. Three patients with bicuspid aortic valves were identified.

A pairwise association analysis was conducted on the clinical data. We report here the main results. P-values are indicated without correction while, with $n = 19$ variables, Bonferoni correction suggests a threshold of $\alpha = 0.05/(19 \cdot 18/2) \approx 2.9 \cdot 10^{-4}$, but the individual analyses were not independent and not of the same importance to our study. Consequently, we also considered a more lenient threshold $\alpha = 0.01$ corresponding to a false discovery rate of 1%. A visualization of the complete pairwise analysis is shown in Supporting information S1 Fig.

The morphological measurements (i.e. aortic diameters and distances of the coronaries from the aortic annulus) were all strongly correlated with Pearson's correlation coefficients generally around 0.5. The lowest correlation coefficients were measured between the diameter of the ascending aorta and the heights of the left and right coronary artery (noted LCA and RCA), respectively $r = +0.20$ for the LCA and $r = +0.13$ for the RCA. The highest correlation

**Table 1. Population demographic, morphometric and echocardiographic description table.** [†]**None-to-severe scale is defined as none/trivial/mild/moderate/severe based on clinical reports.**

|  | Unit | N | Mean (std.dev) or counts | Range |
|---|---|---|---|---|
| Demographics |  |  |  |  |
| Age | year | 130 | 82.0 (7.9) | [54.0 ; 97.0] |
| Sex | male/female | 130 | 63/67 (48.5%) |  |
| BSA | m$^2$ | 107 | 1.81 (0.21) | [1.39 ; 2.35] |
| BMI | kg/m$^2$ | 107 | 27.77 (7.16) | [14.88 ; 54.39] |
| Hypertension | yes/no | 130 | 104/26 (80.0%) |  |
| Coronary disease | yes/no | 130 | 65/65 (50%) |  |
| Euroscore II | % | 118 | 3.9 (6.8) | [0.8 ; 61.0] |
| Frailty score (CSHA) | 1-to-7 | 130 | 3.79 (1.67) | [ 1 ; 7] |
| Aortic diameters |  |  |  |  |
| Annulus (Ann) | mm | 126 | 27.0 (2.4) | [21.3 ; 33.8] |
| Sinus of valsalva (SoV) | mm | 126 | 32.5 (3.5) | [25.0 ; 41.7] |
| Sinotubular junction (StJ) | mm | 126 | 28.6 (3.0) | [21.0 ; 36.2] |
| Ascending aorta (AAo) | mm | 126 | 34.3 (3.8) | [25.1 ; 47.3] |
| Coronary heights |  |  |  |  |
| Right coronary artery (RCA) | mm | 126 | 14.7 (3.2) | [2.0 ; 24.1] |
| Left coronary artery (LCA) | mm | 125 | 13.1 (2.6) | [6.6 ; 19.4] |
| Calcification |  |  |  |  |
| Valve calcium volume | mm$^3$ | 123 | 2,143 (1,135) | [315 ; 5,909] |
| Echocardiographic characteristics |  |  |  |  |
| Stroke volume | mL | 84 | 70.8 (22.4) | [28.0 ; 153.0] |
| Ejection fraction | % | 96 | 54.4 (11.3) | [12.0 ; 77.0] |
| Pressure gradient | mmHg | 130 | 43.6 (12.9) | [ 19 ; 90] |
| Aortic regurgitation | none-to-severe[†] | 92 | 31/0/46/13/2 |  |
|  | % |  | 34%/0%/50%/14%/2% |  |

was observed between the sinus of valsalva (SoV) and the sinotubular junction (StJ) diameters ($r$ = 0.82).

Multicollinearity was observed between sex, aortic diameters and calcification volume. In particular, the SoV and StJ diameters were strongly associated with sex ($p < 1 \cdot 10^{-6}$ for independent t-test) while the association was weaker for the ascending aorta diameter. It was worth noting that, in comparison, correlation between diameters and BSA was much weaker ($r$ = 0.23, $p$ = 0.02 for the SoV for example). Total calcification was on average lower in women (mean 1875 mm³) than in men (mean 2463 mm³), with a t-test p-value of 0.004, but this difference was not significant when correcting for annulus diameter ($p$ = 0.15). The correlation between calcium volume and diameter was always significant and at the strongest for the SoV ($r$ = 0.63, $p < 1 \cdot 10^{-6}$). Calcification was also associated with higher pre-operative pressure gradient accross the valve as estimated for echocardiograms ($r$ = 0.35, $p$ = 5.9 $\cdot$ $10^{-5}$). No meaningful difference in calcium volume was observed in subjects with reported hypertension or coronary disease (only marginally higher in both cases).

Finally, sex, BSA and BMI were positively correlated. Frailty was negatively associated with BMI ($r$ = −0.25, $p$ = 8.0 $\cdot$ $10^{-3}$) but on average lower for men (3.4 versus 4.2 for women, $p$ = 0.008). Coronary disease cases were more frequent in the male group (65% versus 36%, $p$ = 0.0016 for Pearson's chi-squared test). The Euroscore was directly associated to the ejection fraction ($r$ = −0.45, $p$ = 1.0, $\cdot$ $10^{-5}$). Other parameters may be intrinsically linked but appeared to be more weakly correlated (such as EF and SV with a $r$ = 0.26, $p$ = 0.02).

**2.1.2. Shape models.** Independent shape models were estimated for the aortic root and the LVOT. Average morphologies and main modes of variation are presented in Fig 1. An estimate of the mean left ventricle is shown for spatial reference. For the aorta model, the average surface distance between the original data and the shape model reconstruction was 0.30 mm (between 0.18 mm and 0.49 mm for every subject). The mean relative volume error is equal to 0.08%. Points with higher error were found near the leaflet edges but the mean error at every point was consistently lower than 0.74 mm. The reconstruction error in the LVOT model was 0.59 mm (min 0.36 mm, max 1.78 mm), with higher error near the cut between LVOT and LV. The higher error is partially due to the surface smoothing and the lack of clear marker to define the cut between the outflow tract and the LV.

The first three shape modes of the aorta model are shown in Fig 1, they highlight differences in size, aortic diameters, angulation and curvature of the proximal ascending aorta, and sinus shape. The corresponding shape modes for the LVOT showed orientation variability

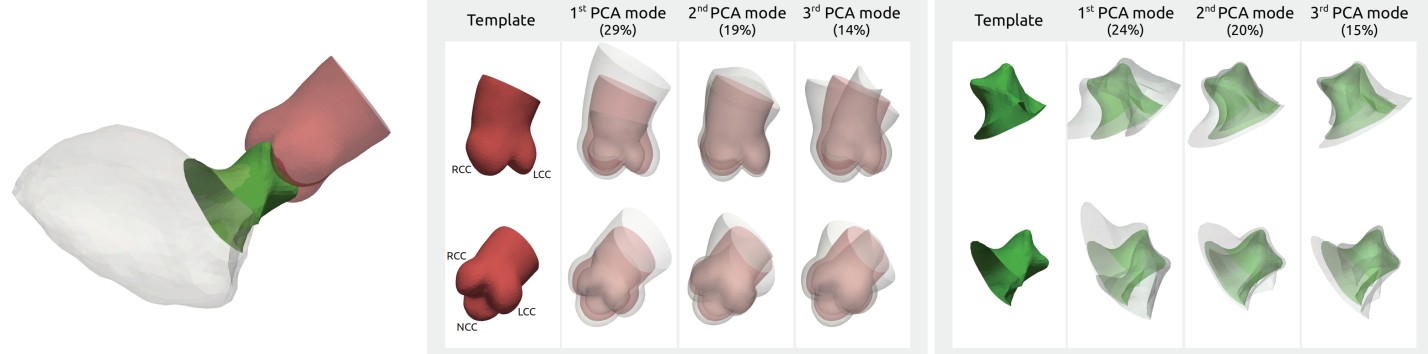

**Fig 1. Shape models.** Left: Templates for the aorta and the LVOT, estimated separately (shaded LV template for orientation reference). Middle: aortic template and first three shape modes. Right: LVOT template and first three shape modes. The transparent grey shapes are shown at -3 and +3 standard deviations.

but features such as diameters, curvatures and length are also observed. These axes illustrated part of the morphological variability in our population without showing any specific feature that would deviate from a normal morphology. The total modelled shape variability measured by the average standard deviation of the vertices positions is equal to 3.67 mm so more than 12 times the reconstruction error (respectively 5.74 mm and almost 10 times for the LVOT model).

**2.1.3. Calcification distribution.** The shape-normalized calcium volume in our cohort varied between no calcification at all and a maximum of 1663 mm$^3$ for one subject (mean 452 mm$^3$, s.d 330 mm$^3$). These measurements correlated with the reported valve calcium volume from CT images ($r = 0.51$) but differences between the two indices, both in term of definition and in term of raw quantification, can be noted. First, the area of interest is not the same: in the 3D analysis the calcification of the aortic wall, and protuberance in the LVOT are modelled, while the clinical measurement focuses on the valve. Second, smaller calcium aggregates may be visible in the images but are not segmented or filtered by the geometrical processing. Finally, shape-normalization does not preserve local volume measurements.

The average calcification map is shown in Fig 2. The calcification rate was higher above the leaflets, mainly near the apposition zones. It was however difficult to distinguish (even on individual cases) if the leaflets were properly fused or if the calcification affected each leaflet separately. The non-coronary cusp (NCC) was on average more calcified than the left-coronary cusp (LCC) and right-coronary cusp (RCC). Measurements based on a cylindrical segmentation of the valve region lead to significant differences ($p < 0.0001$ for the one-way ANOVA) with 152 mm$^3$ (s.d. 130 mm$^3$) for the NCC, 108 mm$^3$ (s.d. 123 mm$^3$) for the LCC, and 90 mm$^3$ (s.d. 92 mm$^3$) for the RCC. The NCC region was larger (respectively 3010 mm$^3$ for the NCC, 2790 mm$^3$ for the RCC, and 2580 mm$^3$ for the LCC) but the relation holds for calcium density. The pattern, disregarding the quantity, was otherwise very similar between

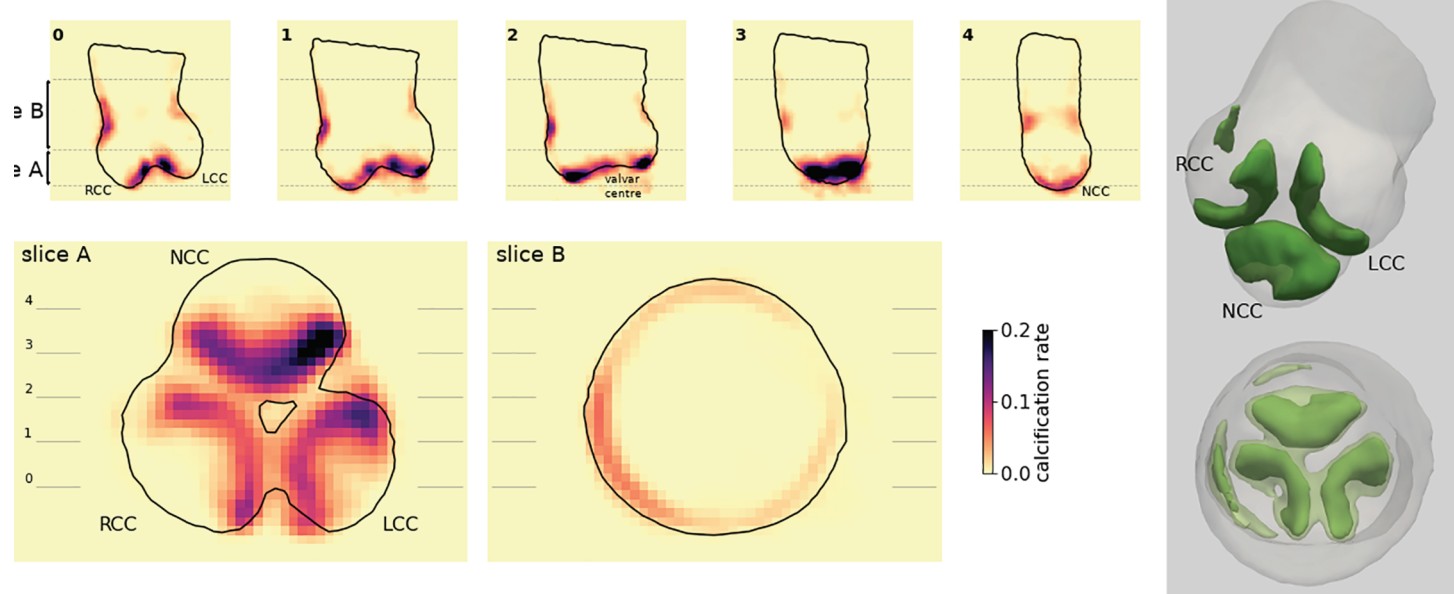

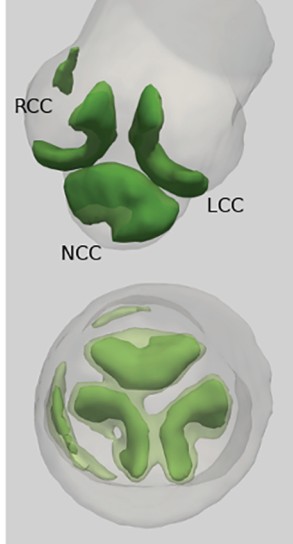

**Fig 2. Average calcium distribution.** Left: five longitudinal cuts (top) and two cross-sectional slices (bottom, A-B). The calcification rate corresponds to the proportion of cases with calcification at this location. The position of the longitudinal cuts are shown on the slice A (on the left), similarly the two cross-sectional slices A and B are shown on cut 0. The cross-sectional slices show the vertical average over the slice. Right: 3D reconstructions with a 10% cutoff (5% cutoff for the transparent surface).

the 3 leaflets: a crescent following the commissure but mainly located on each cusp. Calcification was also visible on the aortic wall (100 mm$^3$, s.d. 173 mm$^3$), with higher concentration on the RCC side. Even if the calcification was highly concentrated around the leaflets and the aortic wall, the combined volume affected by calcification was relatively large (10320 mm$^3$, corresponding to around 40% of the total segmented volume for the aortic root) and the individual calcification patterns were highly variable: only 244 mm$^3$ were commonly calcified in more than 25% of cases, and only 1 mm$^3$ (i.e. one voxel) by at least half the population.

## 2.2. Shape correlations with clinical parameters

### 2.2.1. Aortic root shape.

Potential association between selected clinical variables and the aortic shape model parameters (momenta) were independently tested. Results are presented in Table 2. The strong relationship with aortic diameters is a validation of the methodology but also give a quantitative reference to analyse further results. The effect size is here primarily quantified by the linear regression r$^2$ and, in the case of the individual diameters the value range from 0.07 for the annulus to 0.13 for the StJ; meaning that these single explanatory variables were able to account, respectively, for 7 or 13% of the total variance in the shape model. The aortic shape also correlated with sex, pressure gradient (PG) and ejection fraction (EF), and the reported calcification volume, the shape statistical effects are visualized in Fig 3. Wider and straighter aortas were associated with higher EF, while more compact aortic root with smaller cusps and narrower sino-tubular junction were positively associated with

**Table 2. SSM statistical testing.** The coefficient of determination $r^2$ indicates how much variance of the shape model was explained by each variable. The p-values were computed for the Hotelling's $t^2$ for binary variables (sex, hypertension, coronary disease, AR) and for the likelihood ratio test for continuous variables (all the other ones). 1000 permutations were used.

| | N | Aorta | | LVOT | |
|---|---|---|---|---|---|
| | | $r^2$ (%) | p-value | $r^2$ (%) | p-value |
| Demographics | | | | | |
| Age | 130 | 2.2 % | 0.165 | 0.5 % | 0.185 |
| Sex (M/F) | 63/67 | 3.3 % | **<0.001** | 2.6 % | **<0.001** |
| BSA | 107 | 1.4 % | 0.067 | 0.9 % | 0.057 |
| BMI | 107 | 1.1 % | 0.121 | 1.1 % | 0.076 |
| Hypertension (yes/no) | 104/26 | 0.3 % | 0.986 | 0.3 % | 0.625 |
| Coronary disease (yes/no) | 65/65 | 1.6 % | 0.015 | 1.4 % | 0.083 |
| Euroscore II | 118 | 1.8 % | 0.520 | 0.7 % | 0.578 |
| Frailty score | 130 | 0.7 % | 0.060 | 0.7 % | 0.825 |
| Aortic diameters | | | | | |
| Annulus | 126 | 7.0 % | **<0.001** | 6.2 % | **<0.001** |
| Sinus of valsalva | 126 | 13.1 % | **<0.001** | 6.1 % | **<0.001** |
| Sinotubular junction | 126 | 13.4 % | **<0.001** | 5.0 % | **<0.001** |
| Ascending aorta | 126 | 9.7 % | **<0.001** | 2.0 % | 0.012 |
| Coronary heights | | | | | |
| RCA | 126 | 4.4 % | **<0.001** | 3.6 % | **<0.001** |
| LCA | 125 | 4.3 % | **<0.001** | 3.6 % | **<0.001** |
| Calcification | | | | | |
| Valve calcium volume | 123 | 5.2 % | **<0.001** | 2.8 % | **<0.001** |
| Echocardiographic characteristics | | | | | |
| SV | 84 | 2.4 % | 0.310 | 2.3 % | 0.175 |
| EF | 96 | 0.9 % | **0.001** | 4.3 % | **<0.001** |
| PG | 130 | 1.0 % | **<0.001** | 1.5 % | **<0.001** |
| AR (yes/no) | 61/31 | 1.1 % | 0.896 | 1.5 % | 0.117 |

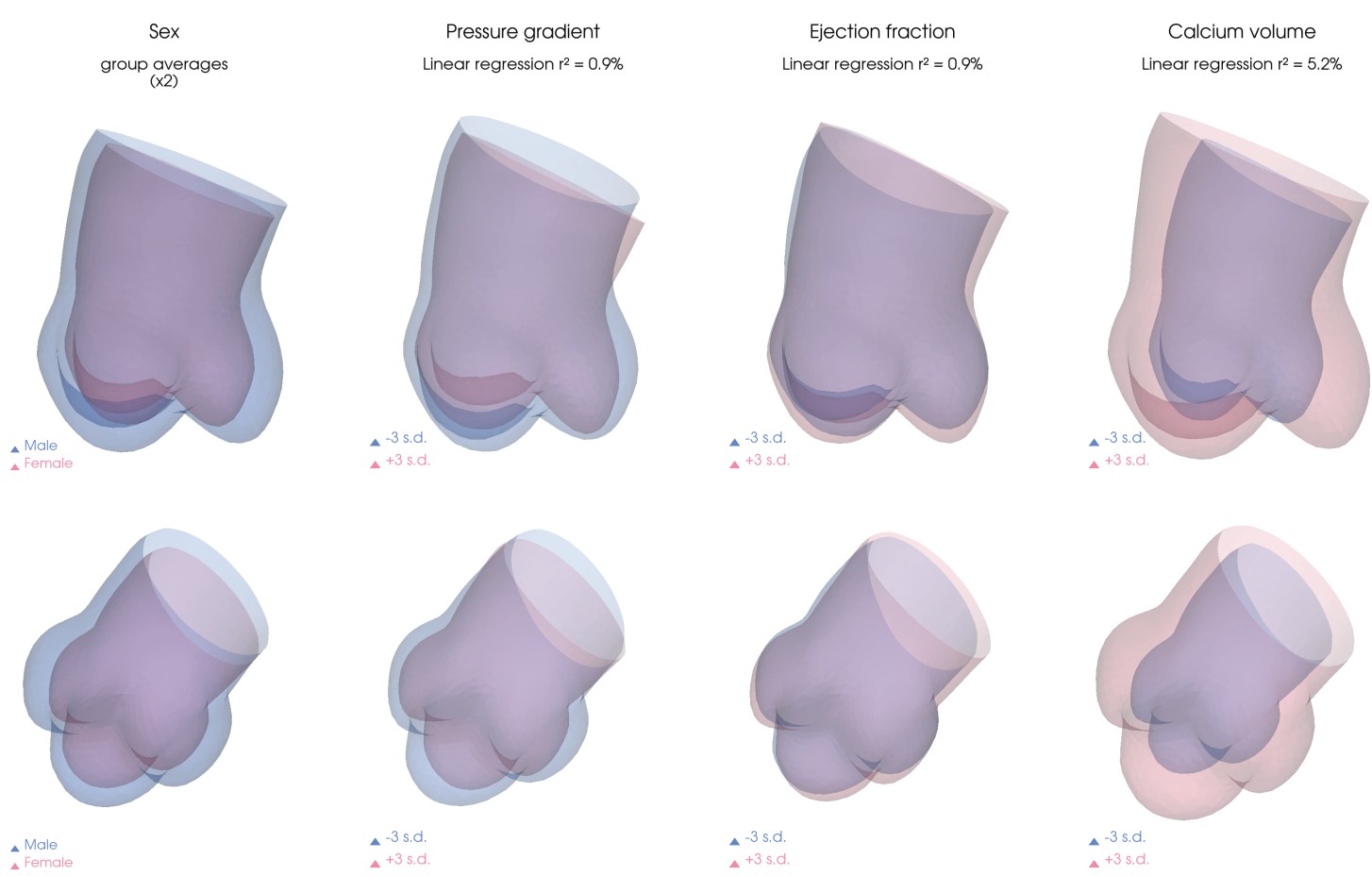

**Fig 3. Mean shapes for sex and linear regression results (at −3/+3 std. dev.) for pressure gradient, ejection fraction and calcium volume.**

higher PG. Weaker (non-significant) associations were also visible with age, BSA and BMI but are not represented.

The relation with sex was mainly caused by size difference but dimorphism was also observed when correcting for (e.g. annulus) diameter (p < 0.01, r2 = 1.9%). More precisely, female average phenotype appeared to be more cylindric while the male average has comparatively larger sinuses and is more curved. We note that when both the annulus and the SoV diameters are included as co-regressor in a multivariate regression the female average is still straighter but the overall effect of sex is highly reduced and not significant anymore (p = 0.4, r2 = 1.6%).

Finally, higher calcification volume was directly correlated to aorta sizes, as seen previously, but the correlation between shape and total calcification was still visible, and significant (p = 0.01, r2 = 2.7%), when corrected for diameter differences: bulkier, straighter aortas with longer and lower sinuses being associated with more calcification.

**2.2.2. Left ventricle outflow track shape.**   For the LVOT, similar correlations were measured with aortic diameters, sex, calcium, and pre-procedural EF and PG. The LVOT shape was also more correlated with BSA than the aorta and preliminary results on the complete LV model showed significant correlation with BSA and BMI. In addition, the analysis after rigid

registration of the LVOT suggested that the realigned shape was less associated with the aortic ascending diameter (p = 0.1), and more with age (p = 0.05).

## 2.3. Factors of calcium distribution

Clinically reported valve calcium volume was correlated with age (p = 0.013), sex (p = 0.004), pressure gradient (p < 0.001), aortic diameters (annulus, SoV, StJ, and AAo with p < 0.001) and position of the coronary arteries (left and right p < 0.001). To better understand how these factors affect the calcium distribution, Fig 4 presents the average calcification and the differences associated with these variables.

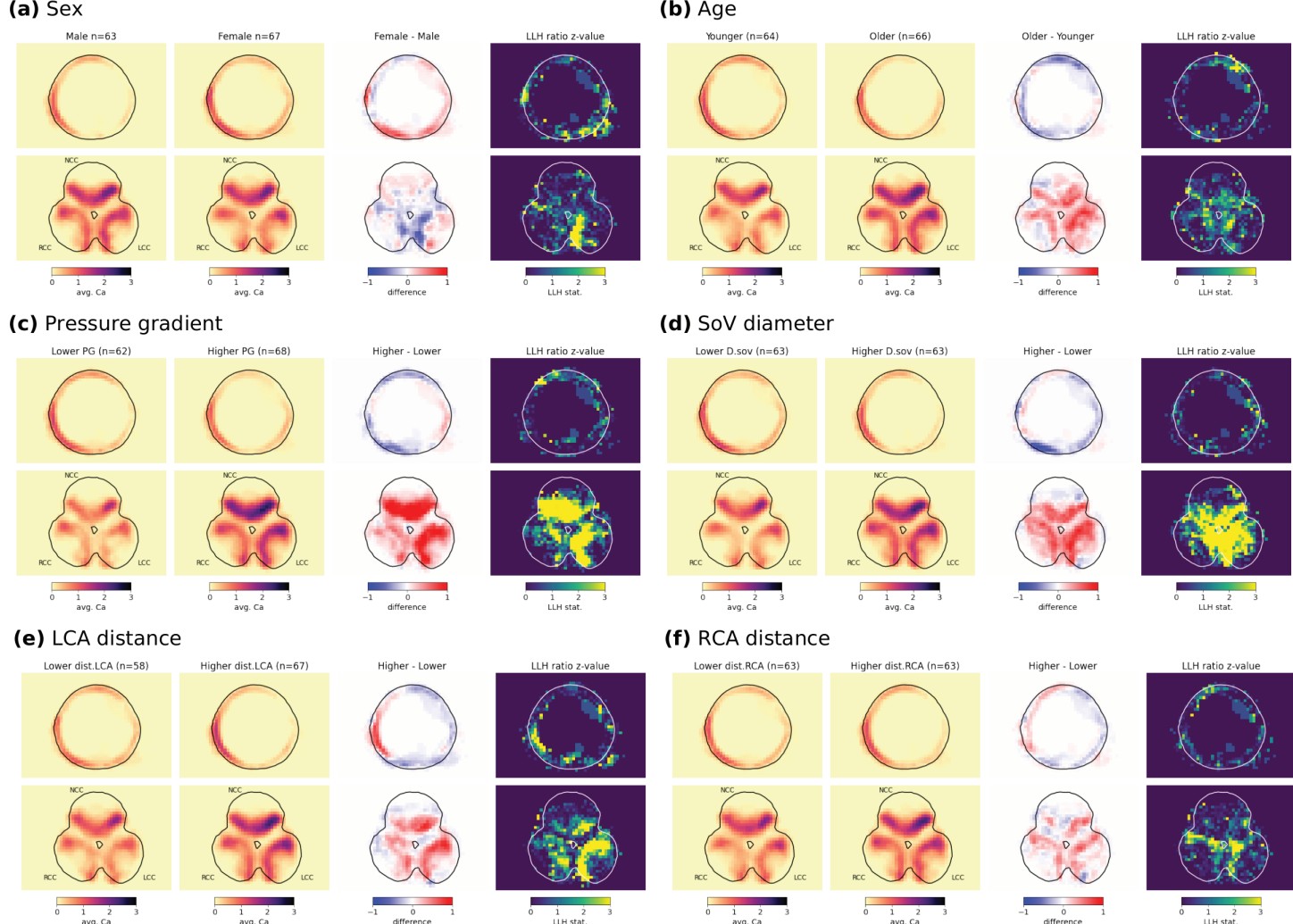

**Fig 4. Calcification rate association with sex, age, and pressure gradient, SoV diameters, and left and right coronary heights.** For continuous variables the population is divided in two groups using the median, the thresholds are: age ≤ 84 y.o., PG ≤ 41 mmHg, SoV diameter ≤ 31.75mm, LCA distance ≤ 13mm, and RCA distance ≤ 14.45mm. Calcification maps are presented for both the ascending aorta slice and the valve slice. For each variable, results are presented from left to right as follow: average calcification rate maps for both groups, difference of rate between the higher and the lower groups, and z-value for the log-likelihood ratio test comparing two distinct populations to the hypothesis of a single one. High z-values, in yellow, indicate areas where the difference is less susceptible to be random, regardless of local calcification rate average and variability.

The significant difference in distribution between the sexes was mainly visible on the RCC and on the LCC with higher calcification visible in males while the level of calcification of the NCC was similar for both sexes. The strongest difference was visible at the commissure between RCC and LCC, possibly an indication of more leaflet fusion between the LCC and the RCC for men. Besides the leaflets, the calcification was on average higher for female in the ascending aorta. The difference is also visualized as 3D reconstruction of the mean calcification for each sex in Supporting information S2 Fig. Younger age was apparently associated with more distributed calcification pattern, including on the walls of the aortic root, while the older patients in the cohort have larger volumes of calcium on the leaflets. The correlation between calcium and PG was strong and affected distinctively the NCC and the LCC leaflets (see Fig 4c).

Regarding morphological measurements, the effect of size on calcification was visible for most measurements, with larger aortas being associated a more calcified valve, and the correlation was the strongest for the SoV diameter. The effect was particularly visible on the LCC and near the commissures, while no differences were visible on the ascending aorta wall (see Fig 4d). The calcification pattern also appeared to be affected by both coronary positions, higher coronary distance being associated with stronger calcification on average, with a stronger effect in the corresponding cusp (see Fig 4e–4f).

## 2.4. Associations with interventional outcomes

**2.4.1. Outcome assessments.** A description of the outcome indicators is reported in Table 3. The average transvalvular pressure gradient dropped from 43.6 (12.9) mmHg pre-operatively to 10.1 (5.0) mmHg post-operatively. The large majority of implanted devices were balloon expandable valves manufactured by Edwards (SAPIEN/SAPIEN 3, n = 110). Ten patients had a permanent pacemaker inserted in relation to the TAVI procedure: 2 in the days before TAVI-procedure, 4 immediately or the day after TAVI surgery, and 4 additional related to the consequences of TAVI. PVL was assessed in 115 cases and classified as either none (n = 30), trivial (n = 23), mild (n = 49), moderate (n = 12) and severe (n = 1). The PVL incidence (mild-to-severe) of 53.9% is comparable to previously published results such as

**Table 3. Peri-operative and outcome indicators table.** *Manufacturers are Edwards Lifesciences, Boston Scientific, Medtronic, and St. Jude Medical. †None-to-severe scale is defined as none, trivial, mild, moderate, and severe based on clinical reports.

|  | Unit | Data avail. | Mean (std.dev) or counts | Range |
|---|---|---|---|---|
| Device |  |  |  |  |
| Device type | balloon/self-expand. | 127 | 110/17 |  |
| Manufacturer | EDW/BSC/MED/SJM* | 127 | 110/8/5/4 |  |
| Device size | mm | 127 | 24.89 ( 2.25) | [20.0 ; 29.0] |
| Conductive disorders |  |  |  |  |
| TAVI-related pacemaker | yes/no | 129 | 10/119 (7.8%) |  |
| Flow indicators |  |  |  |  |
| post-op. PG | mmHg | 111 | 10.14 ( 4.99) | [2.0 ; 29.0] |
| post-op. AR | none-to-severe† | 124 | 35/0/78/10/1 |  |
|  | % |  | 28%/0%/63%/8%/1% |  |
| post-op. PVL | none-to-severe† | 115 | 30/23/49/12/1 |  |
|  | % |  | 26%/20%/43%/10%/1% |  |
| PVL position | R-LCC/other/NCC | 26 | 13/3/10 |  |
|  | % |  | 50%/12%/38% |  |
| 1-year follow up |  |  |  |  |
| Death before 1y | yes/no | 130 | 16/114 (12.3%) |  |

Gilbert et al. [12] reporting a 43.2% pre-discharge PVL incidence for Edward Sapien valves (50.5% for Edward valves only in our population) and of 56.7% for Medtronic core valves.

The position of the PVL was reported in 26 cases and predominantly occurred between the left and right coronary cusps (n = 13) or on the non-coronary cusp (n = 10), as shown in S3 Fig.

**2.4.2. Association with clinical variables.**   The potential associations between demographic and clinical variables and the outcome indicators were explored and: after correction for multiple testing, no significant correlation was observed. In particular, no difference in average calcification score is observed between cases with no PVL (2,362 $\pm$ 1,187 mm$^3$) and cases with mild or more PVL (2,218 $\pm$ 1,115 mm$^3$).

Nevertheless a lower average BMI was noted in the no PVL group compared to the mild and above PVL group (mean 26.1 kg/m$^2$ versus 31.4 kg/m$^2$, t-test p = 0.001) despite no direct link with morphological variables. A different distribution of PVL levels between male and female (p = 0.01 for the Pearson's chi-squared test) was also observed, with higher proportion of no PVL in males and higher proportion of trivial PVL in females as shown in Fig 5. The respective frequencies of no and trivial PVL was 35% and 9% in males and 17% and 31% in females. The other PVL frequencies were more similar (around 43% for mild and 11% for moderate or severe).

A negative association between PVL and device size (t-test p = 0.0003) was observed, with bigger devices associated with less leakage (see Fig 5). Device size was also associated with post-op pressure gradient (p < 0.001) but no direct relation between pressure gradient and PVL was detected.

**2.4.3. Device sizing.**   The choice of device and device size is an important clinical decision that directly impacts the device performance. It can be affected by device availability and by various anatomical characteristics (ellipticity, curvature) or clinical risk factors (of rupture for example) but is first and foremost based on the annulus nominal area. To further study the impact of device sizing, the implanted prosthesis size in relation to the annulus diameters and the different PVL levels and genders were visualized in Fig 6. In order to have comparable measurements, this analysis was restricted to SAPIEN balloon inflatable valves (n = 110). This device is available in four sizes: 20 mm, 23 mm, 26 mm, 29 mm.

The 29 mm device was almost exclusively associated with no PVL and male patients, while the 23 mm device was more often associated with mild PVL and female patients. The difference in outcome between the two devices was significant (p < 0.0001 for the Pearson's

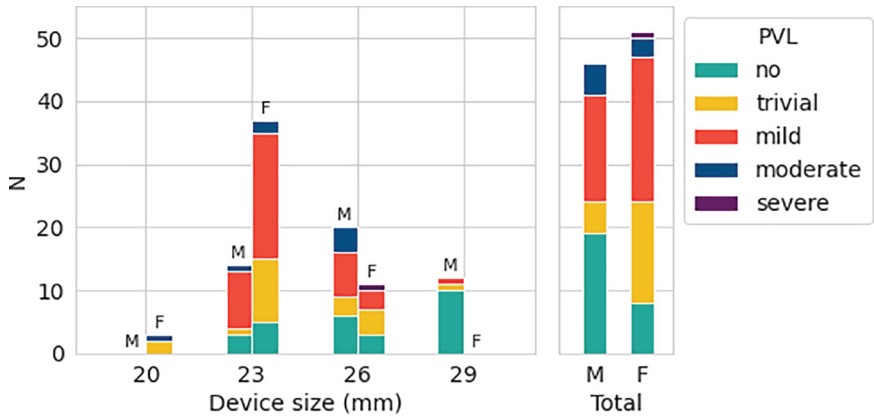

**Fig 5. Histograms for PVL assessment grouped by device size and sex.** SAPIEN device only.

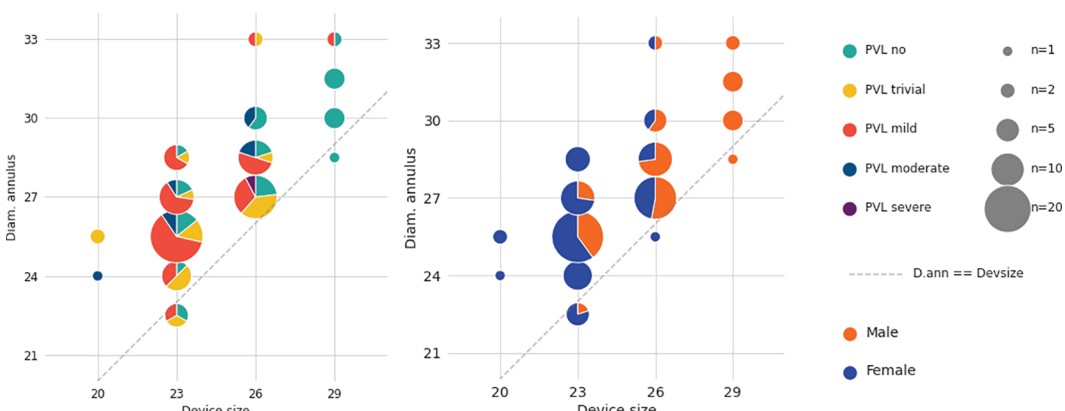

**Fig 6. Impact on device size on PVL (left) and sex distribution on device size (right).** Data was binned per device size and annulus diameters with 1.5 mm intervals.

chi-squared test) and partially explains the differences between the two sexes. Restricting the analysis to the 23 mm and 26 mm devices only, the differences were not as clear but the higher number of trivial cases in the female group was still noticeable (27% in female cases versus only 8% in male cases, p = 0.005).

The range of implanted annulus diameters was relatively lower for the 29 mm device. Normalized for the nominal size, the range were around 11% for the 20 mm, 31% for the 23 mm, 31% for the 26 mm, and 16% for the 29 mm (the standard deviations were non significantly different for the Bartlett's test). In order to assess the relation between sizing and PVL, the distribution of both the annulus and SoV diameters of the cases implanted with a 23 mm or 26 mm devices was looked at. The diameters were normalized by the implanted device size. No significant differences were observed for the annulus but cases with trivial PVL had, in average, a comparatively smaller SoV diameter for their device size (i.e. apparent oversizing, p = 0.04), while mild or more PVL cases had in average a larger SoV diameter (i.e. apparent undersizing, p = 0.04). It is important to note that relation between the annulus and the SoV diameters is different between males and females, females having an average smaller SoV diameter even at comparable annulus diameter as shown in Fig 7. In fact, correcting for both diameters decreases the difference in outcome between male in female (or vice versa, correcting for sex decreases diameter effect). A good separation between the implanted device sizes was generally observed, except for female cases with an annulus diameter between 26 and 29 mm and a SoV diameter below 35 mm that are associated with both 23 and 26 mm devices. It was also observed that the 20 mm device is rarely used, even for smaller cases.

To summarize, morphological dimorphism does not fully explain the differences in device choice between males and females (an apparent small under-sizing in females cases) and leads some situations to be gender specific (medium annulus and small SoV being typically a female phenotype associated with a more variable device choice).

**2.4.4. Calcification and sizing.** The effect of valve calcification on the surgeon's decision regarding device type and size can also be seen in the data analysis. First, self-expandable devices were implanted in patients (n = 17) with lower calcification in average (mean 1770 mm$^3$ versus 2205 mm$^3$, s.d. 1120 mm$^3$, t-test p = 0.15).

Second, focusing on the 23 mm versus 26 mm use-case again, we observed the effect of calcification was visible on the sizing decision. Indeed, less calcification was measured in cases implanted with 26 mm than in the 23 mm cases (2000 mm$^3$ versus 2860 mm$^3$, s.d. 1110 mm$^3$,

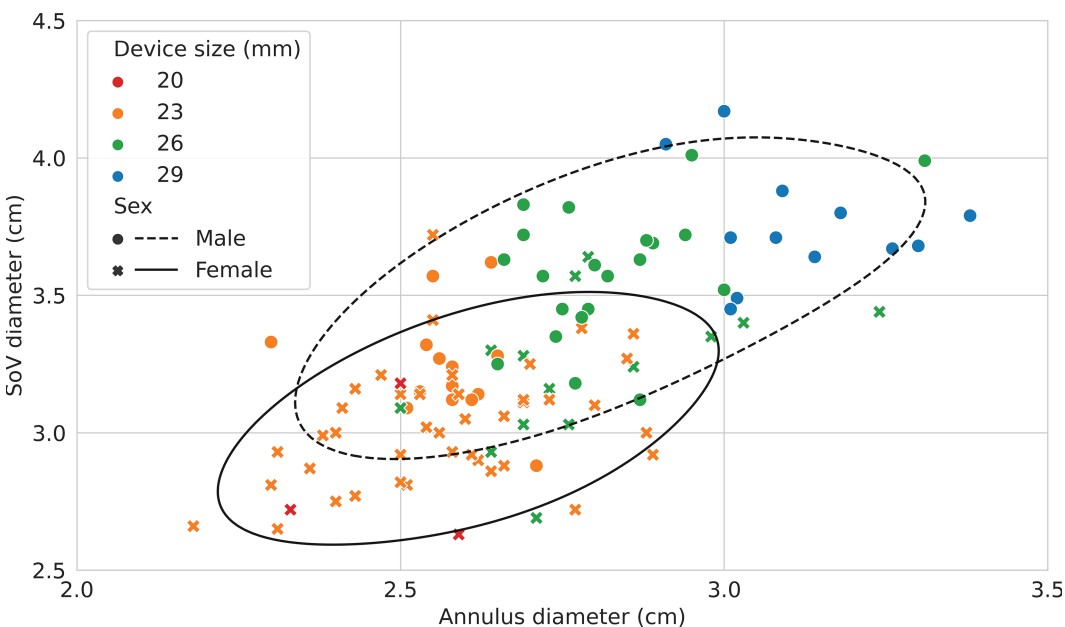

**Fig 7. Difference in size between male and female, and implanted device size.** Even at comparable annulus diameter, female cases have in average a smaller SoV, this leads to different interactions with the selected implanted device. The contours for male and females are thresholded Gaussian kernel density estimates (bw = 2, p = 0.4).

t-test p = 0.002) while the calcification was, as seen previously, positively correlated with the aorta diameters. A logistic regression model, fitted to predict the device choice in function of the annulus diameter, the SoV diameter, the sex and the calcification volume, confirming this result: high calcification weights towards the choice of the smaller device.

This observation naturally questions the results presented in the previous section where smaller devices were, in average, under-performing. Calcification is non-significantly higher in patient with mild or more PVL compared to the no PVL control group (2420 $mm^3$ versus 2000 $mm^3$, s.d. 880 $mm^3$, t-test p = 0.19). The weighting of the calcification towards higher PVL decreased, but stays positive (non-significantly), when sizing variables, such as the relation between annulus and SoV diameters and device size, were included in a logistic model predicting PVL. It suggests that higher calcification leads to more negative results, but part of the effect was caused by the (necessary) clinical choice of a smaller device. The causality relations are however multiple and the analysis does not allow a definitive answer.

**2.4.5. Shape association with outcomes.** Several observation could be made regarding the relation between shape and post-operative indicators. First, a differences in mean aortic morphology was found between cases with no PVL and cases with mild or more PVL. A 4-group categorical model also lead to significant results (see Table 4). On the contrary, a linear model treating the level of PVL as a continuous variable did not seem to represent well the morphological difference associated with a varying PVL severity.

The mean shape of every PVL group is rendered in Fig 8. The main differences are observed in the average diameters of the root with 'trivial' being smaller, the elongation of the LCC in higher PVL levels, and the varying angle of the ascending aorta. The Hotelling's statistic for the comparison between PVL 'none' and 'mild and above' highlights that significant differences are located primarily in the LCC cusp. It also seems that the shape differences is clearer between 'none' and 'mild' than between 'none' and 'moderate', or between 'mild' and

**Table 4. SSM statistical testing for relationship between shape model parameters and clinical post-operative indicators. Likelihood ratio test for continuous variables and categorical (with one-hot coding), and Hotelling's $t^2$ for binary variables, 1000 permutations. Positive PVL is defined as mild or more.**

| | N | Aorta | | LVOT | |
|---|---|---|---|---|---|
| | | $r^2$ (%) | p-value | $r^2$ (%) | p-value |
| Device type | | | | | |
| Balloon/Self-exp. | 110/17 | 2.3 % | 0.058 | 1.2 % | 0.211 |
| Post-operative measurements | | | | | |
| PG | 111 | 0.9 % | 0.565 | 0.5 % | 0.527 |
| PVL | | | | | |
| no/yes (mild or above) | 30/62 | 1.8 % | **0.005** | 0.8 % | 0.453 |
| none-to-severe (categorical) | 115 | 2.9 % | **0.010** | 2.5 % | 0.760 |
| 0-to-3 score (continuous) | 115 | 0.7 % | 0.336 | 0.5 % | 0.729 |
| TAVI-related pace maker | | | | | |
| no/yes | 119/8 | 2.3 % | 0.046 | 1.4 % | 0.224 |

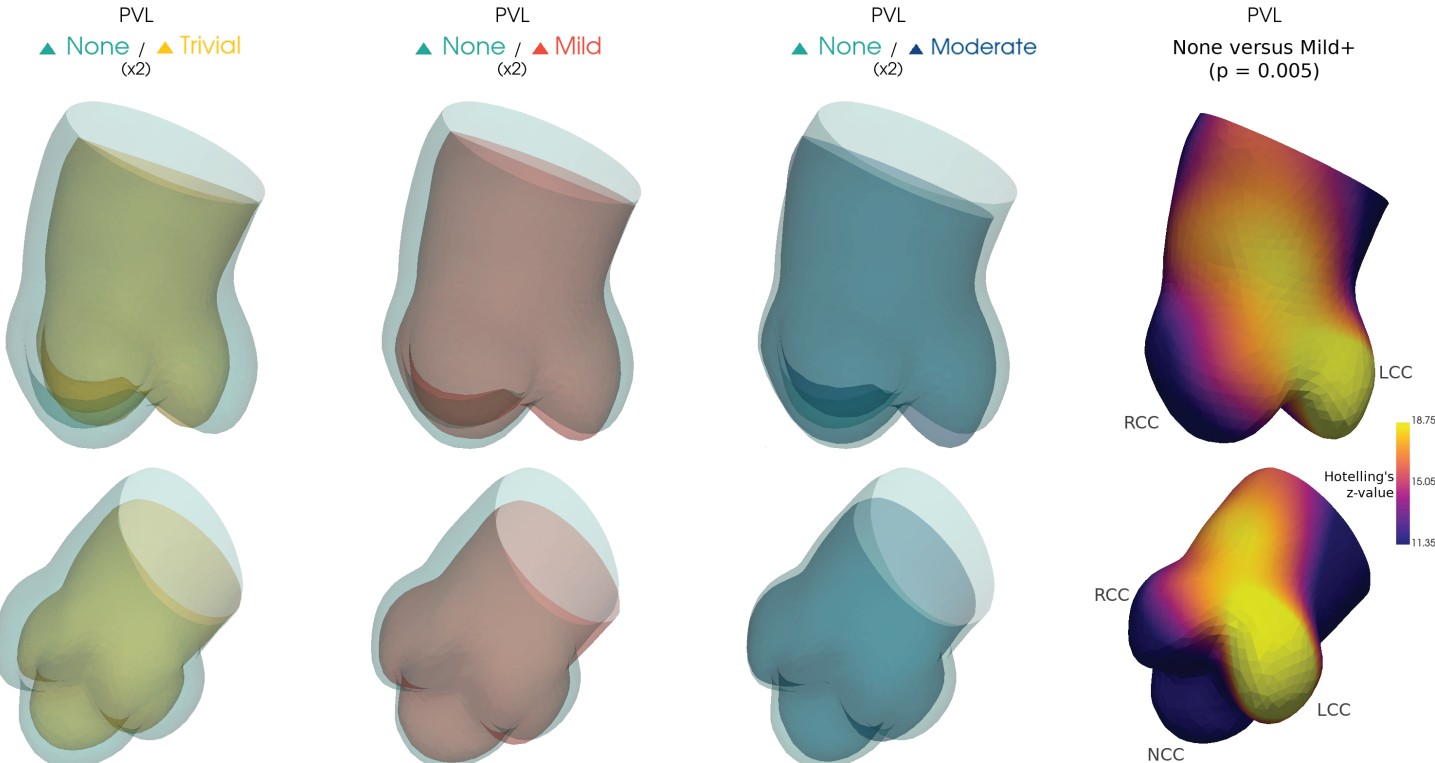

**Fig 8. Mean aortic shape for the different post-op PVL values.** Differences exaggerated x2. Right: Hotelling's statistic for the comparison PVL 'none' (0) vs 'mild and above' (1+), computed using 1000 permutations, the colormap extends from $z_{\alpha=0.5}$ in purple to $z_{\alpha=0.05}$ in yellow.

'moderate'. It was also observed that 'none' and 'trivial' are different when it comes to shape characteristics even if they are often not clinically differentiated due to the low risk of complications following trivial PVL. The non-linearity of the observed changes from none, to trivial, to mild, and finally to moderate reflects the superiority of categorical model over the linear model to represent the shape variations associated with PVL.

To further study the interest of the shape features, we restricted the analysis to the most common device type (SAPIEN model) and included both diameters and the device size as co-regressors. The positive (mild or plus) PVL average still had a longer LCC sinus than the control group, but the difference was small (around 1mm difference) and non-significant (p = 0.5, r2 = 0.9%). Including sex as a co-regressor does not affect this result.

For the left ventricle, the same shape analysis highlighted no significant correlations with PVL. A weak link was nevertheless observed between the realigned LVOT and post-operative PVL (see S4 Fig in Supporting information) with noticeable differences between no PVL and mild PVL or above (p = 0.09).

**2.4.6. Calcification distribution and operative outcomes.** Fig 9 shows three main results: a) the comparison of the average calcification rate between the no or trivial PVL and the mild or above PVL groups, b) the comparison between no pacemaker and TAVI-related TAVI-related, and c) the reconstruction of the average calcification for each PVL level. This spatial analysis did not highlight any clear localized calcification differences associated with operative outcomes, neither PVL nor pacemaker implantation. It is coherent with, and extends the absence of correlation between global calcium volume and PVL level in the cohort.

Variations could be observed in the mean calcification pattern between the different PVL groups. The no PVL average showed three regular crescents, one on each leaflet, while the calcification for trivial PVL was lower on the RCC and LCC but higher on the wall. The average for mild PVL was similar to the control group but the calcification of each leaflet was more condensed, with thicker aggregates that were not connected. Finally the average for moderate

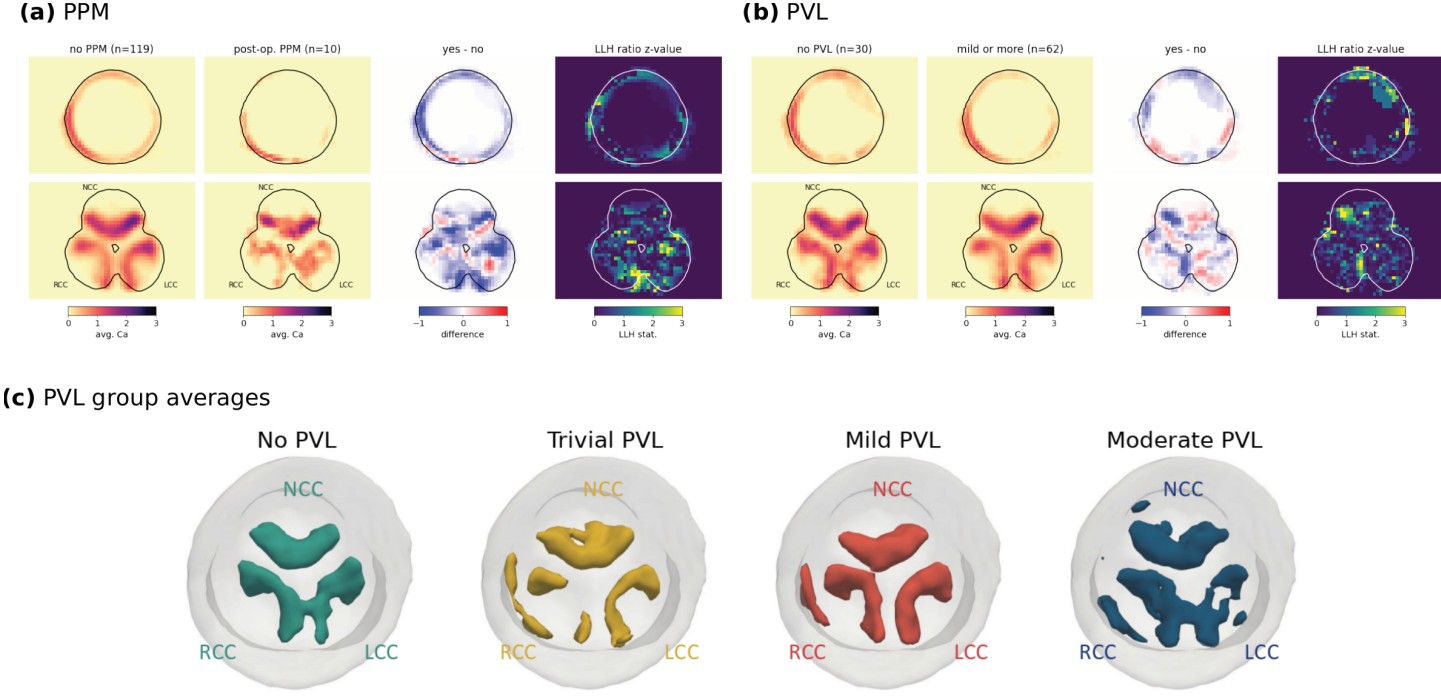

**Fig 9. Relation between calcification and TAVI outcome assessments.** a) Comparison of the average calcification rate between the no or trivial PVL and the mild or above PVL groups. b) Comparison between no pacemaker and TAVI-related pacemaker. c) Reconstruction of the average calcification for each PVL level. For a-b), similar to Fig 4, results are presented from left to right as follow: average calcification rate maps for both groups, difference of rate between the higher and the lower groups, and z-value for the log-likelihood ratio test comparing two distinct populations to the hypothesis of a single one.

PVL was more fragmented, maybe due to the smaller sample size, and cover a larger area in particular on the RCC. Overall the larger variability between the groups seemed to be on the RCC while the NCC was more uniformly calcified. These differences were also non-linear, in the sense that the outcome severity was not associated with monotonic variations of the calcification at a specific position. In particular, it highlighted specific differences between the no PVL and trivial PVL groups. Differences that may not be relevant for the patients clinical follow-up, and which were not observed for higher PVL levels.

## 3. Discussion

Our study presented a complete morphological characterization of the implantation site in patients diagnosed with aortic stenosis who were referred for TAVI procedures. It also explored the relationships between morphological variations and clinical findings both before and after the procedure.

**Morphological description of AS patients.** The proposed 3D non-parametric description allowed to distinguish variations of the aortic root and LVOT in terms of size, diameters, angulation and sinus shape beyond expertly designed scalar parameters [13,22]. The calcification was found to be highly concentrated on the leaflets in crescent areas following the commissures, and was, on average, higher on the NCC compared to the left and right coronary cusps. Such pattern was in agreement with previous observations of higher calcification on the NCC [23–26] but the 3D calcification model, achieved for the first time using shape analysis, allowed for a fine statistical description of the calcification that is not dependent on a difficult valve segmentation.

Such enriched characterization might contribute to a more informed generation of detailed models of AS geometries for in-silico clinical trials - such as the one recently proposed by Verstraeten et al. [21]. Indeed, the SSM of the aorta and LVOT, the associated calcifications, and their relations to demographic and clinical variables, can be used to generate synthetic anatomies that reflect the morphological variability observed in a real patient population, helping both the design and testing of TAVI devices. The identified morphological traits (size, diameters, sinus and LVOT shape, calcium distribution) can also suggest features of importance for procedural planning, stressing common patterns of calcification that could impede proper valve expansion and sealing and would require specific care. Future studies will focus on quantifying the presented relationships and incorporating them into clinical decision-making processes and computational testing.

**Relationship between morphology and demographic data.** Despite the potential age-related remodelling and calcification processes reported in populations with AS [27], no significant morphological correlation with age was observed. We however noted that younger patients (less than 84 y.o.) displayed a more homogeneous calcification pattern, including on the walls of the aortic root, whereas older patients had larger volumes of calcium on the leaflets. As TAVI is increasingly used as a treatment on younger patients, and as calcification is also known to be dependent on inflammation, renal function and phosphate metabolism, and diabetes, this subjective observation may be of interest for future investigation.

Sex emerged as an interesting variable, strongly related to the morphological findings. Male patients showed straighter cylindrical aortas and higher presence of calcium on the RCC and LCC. Total calcification was generally lower in women than in men in accordance with previously reported [28] but this difference was not significant when accounting for diameter differences. The morphological differences and the spatial variation of calcification pattern may contribute to explain the difference in severity and hemodynamic response observed between sexes for identical valve calcification, and complement other identified sex-related

differences such as the level of tissue fibrosis [28–30]. Potential explanations, including hormonal effect or difference in cohorts, of the calcification differences have been proposed [31] but the study of their interactions with other comorbidities, risk factors, disease presentation, diagnosis, and treatment remains a challenging and important research area that suffer from lack of data [31–33].

**Relationship between morphology and pre-operative data.** Aortic diameters were positively correlated with higher calcium volumes reported clinically. The shape analysis provided evidence that the relation between calcium and shape is not only driven by the diameters of the aorta, but also by other features such as the sinuses' shape and length and that, reversely, diameter differences are not associated with uniform calcification changes. For example, the increase of calcium associated with the SoV diameter was particularly prominent on the LCC and near the commissures, while no differences were visible on the aortic wall. Furthermore, a greater coronary distance from the annulus was associated with stronger average calcification on the respective cusp for both coronaries.

The reasons behind these correlations are not fully understood though several hypotheses can be formulated. Selection bias is a first possible explanation: larger valves may tolerate more calcification before being considered severe AS requiring TAVI. Another hypothesis may involve the properties of the blood flow. In this work, we observed both shape and calcification correlations with pre-operative pressure gradient: PG positively correlating with more compact aortic root with smaller cusps, narrow STJ and higher calcification levels. If obstructed valves are logically associated with higher pressure gradient, flow properties could also affect the calcium aggregation through washout problems or higher wall stresses. In particular, it has been suggested that larger valves may be more calcified due to higher leaflet stress [26] or that the different levels of calcification of each leaflet could also be linked to the flows in respective cusp, with the coronary cusps having more washout than the NCC [23]. These mechanisms might also possibly explain the relation between coronary position, disease and calcifications. This could tie in with sex differences in coronary artery disease. While the interaction between aortic shape features and blood flow properties have been studied extensively, the consequences on calcification are more uncharted because of the difficulty to simulate the long term changes of biomechanical properties of the aortic tissues and blood flow [26,27], due to the multiple factors and biological processes involved in calcification proliferation [34] and because of the difficulty to assess the results in-vivo [24,35].

**Relationship between morphology and post-operative paravalvular leakage.** The position of the PVL in the reported cases mainly occurred either between the LCC and RCC or on the diametrically opposite side, next to the NCC, which is in line with findings from studies focusing on this question [12,36]. Our study reports differences of shape and calcification between the different PVL groups. The elongation of left coronary sinus was significantly associated with higher risk of PVL and differences between the no PVL and trivial PVL groups have been observed: the trivial group average was smaller compared to the other groups, especially with a narrow root, and less curved. Calcification distribution and postoperative outcomes of PVL have previously been linked [17] through measurements of the aggregates [18] or by combining multiple local quantification [37] but to the authors' knowledge never though SSM. The exact relation is however unclear and carefully crafted risk scores may not be generalizable without better understanding. For example, some reports found no relation between calcification and PVL [38] and different results are expected depending on the clinical settings, patient selection, implantation approach and, device type and design [17] with improved performance in recent models. In our study, we observed different calcification patterns in each PVL level, with lower calcification in the trivial group, and a slightly higher level of mean calcification between the LCC and RCC in the moderate group. Very

little differences in average calcification levels are visible on the NCC between the different PVL groups. The larger difference between the groups seems instead to be on the RCC. It is, however, uncertain if this could be used as an indicator on an individual level. Morphological features known to impact the TAVI performance would already have been adjusted for, thanks to the operator's knowledge and experience. Their effect will be modulated by the intervention decisions and the perceived likely performance of the different options (device choice, sizing and implantation site). We have, for example, seen how the amount of calcification impacted the choice of device and size with a preference for balloon-inflatable, smaller devices in highly calcified cases in order to limit the intervention risk. Post-operative assessment of the position of the implanted devices and of the dynamic motion of the aortic root would also greatly improve the potential of the study.

Our study also showed how the shape and calcification differences cannot be considered independently to explain the intervention outcomes and the analysis highlighted the entangling of multiple factors such as: shape, calcification, device choice, clinical parameters and gender. We observed differences in outcome between male and female patients that were only partially explained by diameter differences (including the fact that the larger device size showed better performance overall) or average calcification and females were comparatively more often affected by trivial PVL even for similar size and calcium density. We highlighted the possible effect of the SoV and Annulus diameter ratio but other sex-related morphological differences are possibly associated with variation in outcome distribution between male and female. In this case, sexual dimorphism would need to be better taken into account when designing valve prostheses.

**Study limitations.** The conclusions of our study are based on a cohort of patients retrospectively recruited from a single centre which might not fully be representative of the AS population undergoing TAVI. In particular, the focus of this study was not on comparative device performance, or studying the most recent devices improvements; and in this context, there was a heterogeneous distribution of TAVI devices with self-inflating devices poorly represented in our sample (only 17/110, 15.5%). A wider multi-centre study covering more recent devices would provide additional insights.

In addition to this, our study's scope was restricted to examining the relationships between pre-operative data - enriched by shape analysis - and PVL. While post-operative PVL is a validated indicator, it does not fully characterise the long-term consequences of the TAVI [39]. A future step would be to gather 1-year follow-up data regarding long-term PVL, complications, readmission, and quality of life changes. Moreover, the accurate assessment of PVL level and position is very challenging and can be hindered by flawed image collection. In particular, it is a retrospective study and, despite consistently collecting Doppler assessment based on clinical guidelines, the inter-operator variability could be important. Additional information could already be collected for the Doppler images (flow pattern and vessel dynamic), the CTA images (coronary anatomy and fibrotic tissue for example), and the operative reports (valve malpositioning and other complications).

The SSM has its own technical limitations. Statistically, we also considered linear relationship between continuous variables (in Pearson's correlation, and in the likelihood ratio test) and higher order relations may have been missed or underestimated. The representation of the morphology is limited by the segmentation: the inherent simplification of using a simple 2D surface, and the inter-operator variability. In particular, the segmentation of the calcium was challenging because of image artefacts near very bright aggregates necessitating multiple manual inspections and corrections by at least two independent operators. This process was time consuming. In future studies, these limitations are expected to be mitigated through the

development and application of more automated approaches, including deeplearning-based methods specifically trained on this cohort.

The reconstructions were all performed from diastolic images to minimize the effect of cardiac motion and thus improve the visibility of the leaflets. It was however impossible to distinguish leaflets that were fused by calcification, and leaflets that are independently heavily calcified by the commissures. More generally, the dynamic aortic root variation during the cardiac cycle was not studied in this work. Exploring its impact on calcification and TAVI outcome could provide valuable insights. First, tissue stress has an impact on the calcification process [34] and it can be estimated based on the deformation between the best-diastole images and the best-systole images that have also been acquired for every patient. Second, the implanted devices are known to be stiffer than the native tissue and the dynamic motion may be related to PVL or conductive impairments. It should however be highlighted that calcified aortas in aging patient are quite stiff and the motion limited.

The spatial variability of calcification patterns makes it difficult to analyse and compare the underlying calcium distribution. In this work, we compared the local calcification rate at a voxel level in several sub-groups, but this approach does not reflect the calcification genesis mechanisms happening in each individual [23,40]. This study also do not consider the importance of non-calcific fibrotic tissue in AS that has been shown to be related to root morphology [30] and TAVI outcomes [41]. Finally, the relation between calcification and root shape is here simplified by either considering calcium correlations with single morphological measurements (and not the 3D shape model) or shape correlations with total calcification volume. Modelling their interactions in all their complexity and multi-dimensionality would be a natural extension of this work.

## 4. Conclusions

In this work, we presented a novel characterization of a population of aortic stenosis patients who underwent TAVI where the clinical features were enriched by the analysis of the whole morphology. We proposed a new methodology to standardize and compare segmented 3D calcification that could be used in future models and description. The observed average calcification reflected the clinical descriptions including the crescent shapes visible on each leaflet, the higher calcification rate on the non-coronary cusp, and the positive correlation with aortic diameters. This study shows how the sexual dimorphism on both the shape and on the calcification cannot be merely reduced to size differences. Morphological differences were significantly associated with PVL assessments. Larger aortas and shorter LCC sinus were associated with less PVL while female phenotypes, smaller and more conic aortic root, were associated with more PVL. Differences in calcification pattern were found but their potential association with outcomes remains unclear. The new model of AS population could inform futures clinical trials or simulation studies to improve device design and clinical decision process.

## 5. Methods and materials

### 5.1. Data and participants

Inclusion criteria retrospectively included every patient, aged >16 year old, affected by AS that underwent TAVI at Bart's Heart Centre during a defined period of five months in 2018, and that consented to be part of a research study. The patients were selected based on level of available information, and were excluded in case of poor image quality, if the surgery performed was valve-in-valve or if there was a presence of beyond normal malformations or

cardiac implants. All patients underwent routine pre-operative computer tomography angiography (CTA) assessment at Bart's Heart Centre, while pre-operative echocardiography assessment was undertaken either at the Heart Center or at one of the connected clinics. Clinical and peri-operative information was collected from the procedural reports and electronic health records. Immediate post-operative echocardiography was undertaken before discharge. The sample size objective was based on similar studies, number of patients available, and practical considerations (time to collect and segment the data). Clinical data and images from 130 patients were retrospectively included in this study.

Primary outcomes included the implantation of permanent pacemaker related to or caused by the TAVI implantation as reported by the clinical records and the clinical assessment for transvalvular regurgitation (TVR) and paravalvular leakage (PVL) based on the discharge echocardiography reports.

## 5.2. Ethics statement

In compliance with the information standard regarding national data opt-out policy for retrospective data, patient records are automatically included onto the NHS digital registry for scientific research purposes except if the patient decides to opt out, in accordance with the UK Data protection law 15 and Art. 89. Data was obtained by researchers upon application and approval by the NHS Digital Data Access Request Service (DARS). After collection, patient records were de-identified by the removal of direct and indirect identifiers and other sensitive information, as much as research purposes allow. Ethical approval for data access and research protocol was granted by the Barts Health NHS Trust Research Ethics Committee under reference number 296325.

## 5.3. Image processing

CTA images were routinely acquired on a Dual Source 90kVp Siemens SOMATOM Force scanner with contrast agent (Iomeron 400). The acquisition was triggered and best-diastole images were used. The slice thickness in the axial direction was 0.75 mm and the resolution was 0.4x0.4 mm. Semi-automatic segmentation of the aorta, left ventricle, valve leaflets and calcification was performed on the pre-operative CTA images using the Mimics software (Materialise, Leuven, Belgium, version 24). The segmentations of the aorta and the left ventricle only include the inner wall (i.e. the "blood pool"). Triangular surface meshes were then exported for each structure of interest. Meshes were then manually rigidly aligned based on the position of the aortic valve. The aortas were then cut, orthogonally to the centerline, such that the length was equal to 1.5 the root diameter. The left ventricles were smoothed and clipped to focus on the implantation space in the left ventricular outflow track (LVOT) and to reduce the variability caused by trabeculations and ventricular shape and position. The cut was defined by a two step process. First, a rough shape model of the whole ventricle was estimated. Then, this model was used to propagate a manually defined planar cut to each individual mesh.

Calcifications at the implantation site were segmented by a semi-automatic process consisting of initial image thresholding to identify the calcified area followed by manual refinements performed on a slice-by-slice basis. Thresholding was adjusted between 300 and 1000 Hounsfield unit, depending on the characteristics of each dataset. To maintain consistency across segmentations, these were performed by a single, trained operator and subsequently reviewed by an experienced member of the research team. The resulting segmentations provided a detailed spatial representation of the calcification distribution and enabled quantitative assessment of calcification volumes. Valve calcium volumes and Agatson's score were

also available in the clinical reports based on an independent processing of the images for clinical follow-up.

## 5.4. Statistical shape modelling (SSM) framework

The shape analysis presented in this work relies on the Large Diffeomorphic Deformation Metric Mapping framework implemented in Deformetrica [42]. In this framework, the simplest model to describe a population of 3D objects (such as surface meshes) is called atlas and is constituted of one template shape, representative of the mean shape, and of a set of smooth and regular deformations that transform this template toward each subject specific morphology.

The space of valid deformations is parametrized by infinitesimal displacements, called momenta, that are defined in the whole 3D space and belong to a reproducing kernel Hilbert space (RKHS) built using a Gaussian kernel with fixed width and a set of control points uniformly sampled in the ambient space occupied by the 3D surfaces. By construction, the deformations are diffeomorphic, fully parametrized by a limited number of spatially localized 3D vectors, and linear operations, and usual Euclidean statistics, can be used in this deformation parameter space. To summarize:

- A shape atlas is a model constituted by a template shape $T$ and by individual deformations $(\phi_i)$.
- The deformations of the template $\phi_i \cdot T$ are approximation of the original subject meshes $S_i$ with the same mesh topology as $T$. The resolution $\lambda_W$ of the current distances controls the scale at which the objects are described and a noise level $\lambda$ is set to balance between the fit accuracy and the deformation regularity.
- Each deformation $\phi_i$ is parametrized by a set of momenta vector $a_k^i(0)$ defined on a common set of control points $c_k$ sampling the 3D space. The scale $\lambda_V$, of the RKHS deformation kernel $K$, controls the stiffness of the deformations.
- Finally, the position of the control points and the initialization of the template can have an impact on the optimized solution.

The choice of the kernel widths, and of a good initialization, can be challenging. Heuristics have been published to guide this process [43] but the decisions often remain subjective. To circumvent this issue, we rely on the assessment of the reconstruction accuracy to guarantee that the model was able to capture the morphological variability in the population and that the optimization was successful. In this work, experiments were first conducted using different kernel sizes (from 5 mm to 23 mm) on a random subsample of the population and a fixed number of iterations. The deformation kernel width was selected based on the maximization of the model's likelihood (after visual inspection of the result to check that no subject was showing aberrant results). The shape kernel width was selected to minimize the surface reconstruction error. The selected parameters were $\lambda_W$ = 7.0 mm, $\lambda_V$ = 11.0 mm, and $\lambda$ = 2.0 for the aorta, and $\lambda_W$ = 5.0 mm, $\lambda_V$ = 11.0 mm, and $\lambda$ = 2.0 for the left ventricle. Control points were distributed in a regular grid over the joint bounding box that include every geometry, with 1 mm margin.

## 5.5. Spatial normalization of calcifications

The aorta shape model was used to normalize the individual calcium distribution. In order to have spatial correspondence between every subject, the deformations template-subject were

inverted and applied to the calcium segmentations: $C_i' = \phi_i^{-1} \cdot C_i$. The segmentations were then subsampled to 3D 1x1x1mm images.

A vertical direction was manually aligned with the aortic root template main axis and used to defined two 2D cross-sectional slices. The first slice was designed around the aortic valve to include the complete leaflets and their calcifications (slice A on Fig 2), the second one was designed above the first slice to include most of the calcification on the ascending aortic wall (see slice B on Fig 2). The 3D calcification maps were then vertically averaged along in this direction for each slice. As a result, slice A was 10 mm thick, and slice B was 20 mm thick.

The shape-normalization induced a volume correction on the calcification. It can be seen as using calcium density, except that instead of using some global measurements (such as the annulus diameter for example) it relies on local shape differences.

## 5.6. Statistical analysis

**5.6.1. Demographic data.** Variables of interest were selected among the reported information according to clinical interest and expert knowledge. No data imputation was done and correlations were computed by excluding cases with incomplete information. The association between two binary variables was estimated using Pearson's chi-squared test for contingency tables. A t-test was performed to compare the mean of a continuous variable in two groups. Finally the Pearson's correlation coefficient was computed to measure the relation between two continuous variables.

PVL was reported as a categorical variable with 5 levels: none, trivial, mild, moderate, and severe. It has been both treated as a binary variable (none and trivial versus mild, moderate and severe) and as a continuous variable by mapping none to 0, trivial to 0.5, mild to 1, moderate to 2, and severe to 3. This choice is arbitrary and the relationships with variables of interest may not even be monotonic.

**5.6.2. Shape model analysis.** The shape of subject $i$ is fully parameterized by a limited number of spatially localized 3D vectors (the momenta vectors $\{a_k^i(0)\}_k$) and linear operations can be used in this deformation parameter space. In particular, it is possible to compute the mean or standard deviation in one subgroup, or to estimate a regression model using a selection of explanatory variables.

Statistical analysis may still be challenging due to the high dimensionality of the data as it is common to have a few hundreds control points. In our case, the problem arised for hypothesis testing. The selected approach [44] combines the estimation of the usual test statistic (a two-group Student's $t$ for example) at every control point with a summary statistic and a permutation scheme to evaluate the hypothesis globally. This approach enables correction for multiple comparison with limited power-loss and require limited hypotheses on both the data distribution at each control point and the spatial correlation. The summary statistic is chosen as the maximum of the test statistic over the control points to control for the family-wise error. In this work, two situations were considered: the assessment of a difference in mean between two subgroups, and the assessment of a linear effect associated with one or several continuous variables (binary variables can, in this case, be treated as continuous) In the first case, the Hotelling's $t^2$ was used as test statistic. In the second case, the likelihood ratio statistic for the linear least-square model was selected.

**5.6.3. Calcification modelling.** We focused in this study on the comparison of the local average calcification rate between different group of patients with the calcification for each patient being characterized by the two 2D slices defined previously. We modeled the probability distribution of the calcium quantification at each point by the product of a Gamma distribution and of a Bernoulli distribution to reflect the measurement of a rate with a high

probability of being null. The scale $\theta$ of the Gamma distribution was fixed ($\theta$ = 2.2) based on preliminary experiments. It reflects the fact that the *scale* of the calcium measurements is identical at every voxel. We also observed that variation in $\theta$ were not affecting the maximum likelihood (i.e. the model fit) as much (see Supporting information S1 Appendix for more details); the Bernoulli parameter $p$ and the shape parameters $k$ depended on the position and selected group of patients.

For two groups of patients $G_a$ and $G_b$, it was then possible to compute the maximum-likelihood ratio to compare the hypothesis $H_0 = \{(p_a = p_b) \wedge (k_a = k_b)\}$ of an identical distribution in both groups to the hypothesis $H_1 = \{(p_a \neq p_b) \vee (k_a \neq k_b)\}$ of two different groups. We denote $(x_i), i \in [1, n]$ the calcium quantifications, at the selected voxel, for $n$ subjects, and let $n_0 = \#\{x_i = 0\}$ the number of cases with no visible calcium at this point. The maximum estimators or $p$ and $k$ are explicit:

$$\hat{p} = 1 - \frac{n_0}{n} \text{ and } \hat{k} = \frac{1}{\theta} \frac{1}{n - n_0} \sum_{x_i \neq 0} x_i$$

The log-likelihood then writes:

$$L(\{x_i\}) = n_0 \log(1 - \hat{p}) + (n - n_0) \log(\hat{p}) + \sum_{x_i \neq 0} \log g_{\hat{k},\theta}(x_i)]$$

where $g_{k,\theta}$ is the probability density function of the Gamma distribution $\Gamma(k, \theta)$. We can then use the logarithm of the ratio as a statistic to locally assess the differences between the two groups:

$$z = L(\{x_i\}_{i \in G_a}) + L(\{x_i\}_{i \in G_b}) - L(\{x_i\}_{i \in G_a \cup G_b})$$

## 5.7. Shape model estimation

Our analysis focuses on the aortic root and left ventricular outflow tract (LVOT) shapes to characterize the morphological variability of the TAVI landing zone (LZ) and its surroundings. The aorta and the LVOT models were estimated independently.

The optimization of a shape model (fitting the template and the individual deformation) is handled by Deformetrica as a regular, non-linear, unconstrained optimization problem using BFGS algorithm and automatic gradient differentiation. The optimization settings and initialization can however have a large impact on the obtained results and it is necessary to first select the hyperparameters of the model (mainly the resolution $\lambda_W$ and the scale $\lambda_V$).

A range of reasonable parameter values for $\lambda_W$ and $\lambda_V$ was first intuited based on the dimension of the studied objects, the size of their finer interesting details and the resolution of the reconstructions. Further optimization was then done using preliminary model estimations based the Akaike information criterion (AIC) for $\lambda_V$ and the average reconstruction error for $\lambda_W$. For the aortic root model, we considered the regularly sampled values between 3 and 15 mm and we finally selected $\lambda_V$ = 11mm and $\lambda_W$ = 7mm. The selected values for the LVOT model were $\lambda_V$ = 11mm and $\lambda_W$ = 5mm.

Finally, a multi-step optimization procedure was conducted to avoid local minima and poorly defined optimization problems when the geometries are initially too different from each other. First iterations are designed to reliably provide an approximate solution that would be a better initialization of the final optimization. The key steps are:

1. Estimation of an approximate template using a subset of subjects, and manually remeshing it with an adaptive vertex density.
2. Initialization of the momenta for every subject using the low-resolution meshes and deformations.
3. It is also possible to fix the template to optimize the momenta, decoupling the optimization for every subject, greatly reducing the problem dimensionality and helping with the optimization convergence.
4. The mean momenta can be used to re-center the template before optimizing the model again.
5. The last two steps may be repeated (generally only once or twice) until convergence is observed.

Accuracy of the final shape model was then assessed by computing the distance between model reconstruction and original 3D surfaces.

## Supporting information

**S1 Fig. Correlation matrix of the pairwise analysis of the clinical data.** Variables are divided between binary and continuous variables. The relation between two variables was assessed by respectively: a Pearson's chi-squared test for two binary variables, a t-test for one binary and one continuous variables, and a Pearson's correlation test for two continuous variables.
(PDF)

**S2 Fig. Mean calcium distribution**. 3D visualization of the average calcium aggregates for the whole population and both genders separately.
(PNG)

**S3 Fig. PVL position chart. Cumulative distribution of reported leakage.** Visualization of the paravalvular leakage position for the 26 cases where it has been reported in our population.
(PNG)

**S4 Fig. Left ventricle in *systole* for each PVL group.**
(PNG)

**S1 Appendix. Calcification density Bayesian modelling.** Probabilistic modelling of the local calcium distribution and comparison to empirical results.
(PDF)

## Acknowledgments

We would like to thank all the collaborating partners within the Simcor project for the fruitful discussions.

## Author contributions

**Conceptualization:** Ebba Montgomery-Liljeroth, Silvia Schievano, Jan Brüning, Wouter Huberts, Claudio Capelli.

**Data curation:** Ebba Montgomery-Liljeroth, Yaxi Chen, Anthony Mathur, Kush Patel, Claudio Capelli.

**Formal analysis:** Raphael Sivera.

**Funding acquisition:** Silvia Schievano, Jan Brüning, Wouter Huberts, Claudio Capelli.

**Investigation:** Raphael Sivera, Ebba Montgomery-Liljeroth, Andrew Cook, Kush Patel, Claudio Capelli.

**Methodology:** Raphael Sivera, Ebba Montgomery-Liljeroth, Claudio Capelli.

**Project administration:** Claudio Capelli.

**Software:** Raphael Sivera.

**Supervision:** Claudio Capelli.

**Validation:** Ebba Montgomery-Liljeroth.

**Visualization:** Raphael Sivera, Ebba Montgomery-Liljeroth.

**Writing – original draft:** Raphael Sivera, Ebba Montgomery-Liljeroth, Claudio Capelli.

**Writing – review & editing:** Raphael Sivera, Ebba Montgomery-Liljeroth, Silvia Schievano, Jan Brüning, Wouter Huberts, Anthony Mathur, Andrew Cook, Kush Patel, Claudio Capelli.

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
