## [Decision Letter · Decision Letter 0]

4 Feb 2025

PDIG-D-24-00265Enriching the characterization of patients undergoing TAVI: what can the aortic morphology and calcification tell us?PLOS Digital Health Dear Dr. Sivera, Thank you for submitting your manuscript to PLOS Digital Health. After careful consideration, we feel that it has merit but does not fully meet PLOS Digital Health's publication criteria as it currently stands. Therefore, we invite you to submit a revised version of the manuscript that addresses the points raised during the review process. Please submit your revised manuscript within 60 days Apr 05 2025 11:59PM. If you will need more time than this to complete your revisions, please reply to this message or contact the journal office at digitalhealth@plos.org. Please include the following items when submitting your revised manuscript:* A rebuttal letter that responds to each point raised by the editor and reviewer(s). You should upload this letter as a separate file labeled 'Response to Reviewers'. This file does not need to include responses to any formatting updates and technical items listed in the 'Journal Requirements' section below.* A marked-up copy of your manuscript that highlights changes made to the original version. You should upload this as a separate file labeled 'Revised Manuscript with Track Changes'.* An unmarked version of your revised paper without tracked changes. You should upload this as a separate file labeled 'Manuscript'. If you would like to make changes to your financial disclosure, competing interests statement, or data availability statement, please make these updates within the submission form at the time of resubmission. Guidelines for resubmitting your figure files are available below the reviewer comments at the end of this letter. We look forward to receiving your revised manuscript. Kind regards, Amara TariqAcademic EditorPLOS Digital Health Amara TariqAcademic EditorPLOS Digital Health Leo Anthony CeliEditor-in-ChiefPLOS Digital Healthorcid.org/0000-0001-6712-6626 **Journal Requirements:**

1. We ask that a manuscript source file is provided at Revision. Please upload your manuscript file as a .doc, .docx, .rtf or .tex.

2. In the online submission form, you indicated that "Access to anonymized data (clinical data and 3D models) can be granted upon request.". 

a. In a public repository, 

b. Within the manuscript itself, or 

c. Uploaded as supplementary information.

 **Additional Editor Comments (if provided):** Reviewers have raised critical concerns about lack of clarity in terms of description of the methodology. Responding to these concerns will improve manuscript quality as well as potential for reproducibility of this work. Authors are advised to revise the manuscript considering reviewers' comment.**Reviewers' Comments:** Reviewer's Responses to Questions

**Comments to the Author**

1. Does this manuscript meet PLOS Digital Health’s publication criteria? Is the manuscript technically sound, and do the data support the conclusions? The manuscript must describe methodologically and ethically rigorous research with conclusions that are appropriately drawn based on the data presented.

Reviewer #1: Partly

Reviewer #2: Yes

2. Has the statistical analysis been performed appropriately and rigorously?

Reviewer #1: Yes

Reviewer #2: Yes

3. Have the authors made all data underlying the findings in their manuscript fully available (please refer to the Data Availability Statement at the start of the manuscript PDF file)?

Reviewer #1: No

Reviewer #2: Yes

4. Is the manuscript presented in an intelligible fashion and written in standard English?

Reviewer #1: Yes

Reviewer #2: Yes

5. Review Comments to the Author

Reviewer #1: Minor Issues

1. Title Improvement

o Recommendation: Consider a more descriptive title, such as: "Morphological and Calcification Predictors of Outcomes in TAVI Patients: A 3D Statistical Shape Modeling Study." This better conveys the focus on outcomes and the modeling approach.

2. Detailed Methods for Statistical Shape Modeling (SSM)

o Issue: The process for selecting control points and kernel parameters in SSM is not fully explained, potentially impacting reproducibility.

o Recommendation: Include additional detail on parameter choices and optimization steps within the Methods section or in supplementary materials.

3. Expand on Sexual Dimorphism

o Recommendation: Sexual dimorphism is discussed, but the underlying mechanisms remain speculative. If possible, explore correlations with specific biological factors (e.g., hormone levels or tissue properties) or acknowledge this as an area for further research.

4. Clarify Use of Open Science Tools

o Issue: Open science practices are briefly mentioned but could be expanded.

o Recommendation: List specific repositories or platforms where data could be hosted (e.g., Dryad, OSF) and provide a rationale for data sharing practices.

Reviewer #2: In the present, Sivera and co-workers sought to enrich the characterization of patients undergoing transcatheter aortic valve implantation (TAVI) by investigating the association between advanced image-based morphological markers and clinical TAVI findings. Primary outcomes (endpoints) included the implantation of permanent pacemaker, assessment of transvalvular regurgitation (TVR) and occurrence of paravalvular leakage (PVL).

CTA imaging was acquired and processed at best-diastole. The lack of CTA imaging at best-systole should be acknowledged as a limitation of the study. Nonetheless, exploring the impact of dynamic aortic root variations between systole and diastole on the presented analysis could provide valuable insights. The authors are encouraged to elaborate further and discuss this aspect.

The authors are to be commended for their transparency in reporting and summarizing the limited impact of SSM-based predictors on TAVI procedural outcomes, such as PVL and pacemaker implantation. From a clinical perspective, the authors are encouraged to elaborate on whether further efforts or future developments are underway or planned to advance preliminary evidence of this research.

Section 5.3: “A base threshold of 300 Hounsfield unit was adapted to each image based on the operator assessment. We stress the fact that the focus of the study was on the spatial description of the calcification pattern and that better segmentations were favoured over consistent estimation of calcification scores.”

Please clarify if CTA imaging was acquired with or without use of contrast agent, given its impact on voxels’ HU intensity.

For the sake of reproducibility, the authors should clarify what is meant by 'better segmentations were favoured' and provide details on how manual threshold corrections were managed, including whether intra- and inter-operator variability was assessed.

Section 5.5: “A vertical direction was manually aligned with the aortic root template main axis and used to defined two 2D cross-sectional slices. The first slice is 10mm-thick around the aortic valve (slice A on Figure 12), the second slice (slice B on Figure 12) is 20mm-thick above the first slice. The 3D calcification maps were then vertically averaged along in this direction for each slice.

The reported paragraph requires further attention. Neither slice A nor slice B is visible in Figure 12, which needs to be addressed. Additionally, the rationale for using 10mm-thick and 20mm-thick planes should be clarified to ensure the reader understands their scope.

Were patients with bicuspid aortic valve (BAV) enrolled in the study?

A total of 130 patients were enrolled. How did the authors determine whether this sample size was sufficient for the analysis using statistical shape analysis?

How “the presented morphological characterization […] could contribute to […] plan future improvement of TAVI device design” (as highlighted at the end of the abstract) should be made clearer to the reader.

Table 3: The authors are encouraged to comment on the rates reported for mild post-TAVI AR (78/124, 62.9%) and mild post-TAVI PVL (49/115, 42.6%). Including the percentage values directly in the table is also highly recommended for clarity and ease of interpretation.

In the discussion, the authors highlight the novelty of the study in terms of the morphological characterization of the implantation site in patients undergoing TAVI (as well as in the conclusion, i.e., “new methodology to standardize and compare segmented 3D calcification”). While the potential of the present analysis is acknowledged, the authors should further elaborate on and critically discuss the novelty of their approach, particularly in the context of prior research focusing on the pre-TAVI assessment of aortic anatomy.

Page 14: “We have, for example, seen how the amount of calcification impacted the choice of device and size with a preference for balloon-inflatable, smaller devices in highly calcified cases in order to limit the intervention risk.” This confirms well-known evidence from previous clinical practice. It is worth noting that only a small cohort of patients (17/110, 15.5%) in this study was implanted with and investigated using self-expandable devices.

6. PLOS authors have the option to publish the peer review history of their article (what does this mean?). If published, this will include your full peer review and any attached files.

**Do you want your identity to be public for this peer review?** For information about this choice, including consent withdrawal, please see our Privacy Policy.

Reviewer #1: No

Reviewer #2: No

---

## [Decision Letter · Decision Letter 1]

3 Jun 2025

Morphology and calcification characterization in patients undergoing TAVI: A 3D Statistical Shape Modelling Study

PDIG-D-24-00265R1

Dear Dr Sivera,

We are pleased to inform you that your manuscript 'Morphology and calcification characterization in patients undergoing TAVI: A 3D Statistical Shape Modelling Study' has been provisionally accepted for publication in PLOS Digital Health.

Best regards,

Amara Tariq

Academic Editor

PLOS Digital Health

**Additional Editor Comments (if provided):**

Authors have satisfied all concerns of the reviewers.

**Reviewer Comments (if any, and for reference):**

Reviewer's Responses to Questions

**Comments to the Author**

1. If the authors have adequately addressed your comments raised in a previous round of review and you feel that this manuscript is now acceptable for publication, you may indicate that here to bypass the “Comments to the Author” section, enter your conflict of interest statement in the “Confidential to Editor” section, and submit your "Accept" recommendation.

Reviewer #2: All comments have been addressed

2. Does this manuscript meet PLOS Digital Health’s publication criteria? Is the manuscript technically sound, and do the data support the conclusions? The manuscript must describe methodologically and ethically rigorous research with conclusions that are appropriately drawn based on the data presented.

Reviewer #2: Yes

3. Has the statistical analysis been performed appropriately and rigorously?

Reviewer #2: Yes

4. Have the authors made all data underlying the findings in their manuscript fully available (please refer to the Data Availability Statement at the start of the manuscript PDF file)?

Reviewer #2: Yes

5. Is the manuscript presented in an intelligible fashion and written in standard English?

Reviewer #2: Yes

6. Review Comments to the Author

Reviewer #2: (No Response)

7. PLOS authors have the option to publish the peer review history of their article (what does this mean?). If published, this will include your full peer review and any attached files.

**Do you want your identity to be public for this peer review?** For information about this choice, including consent withdrawal, please see our Privacy Policy.

Reviewer #2: No
